# Synthesis of 2,2,6-Trisubstituted 5-Methylidene-tetrahydropyran-4-ones with Anticancer Activity [note 1]

**DOI:** 10.3390/molecules25030611

**Published:** 2020-01-30

**Authors:** Tomasz Bartosik, Jacek Kędzia, Joanna Drogosz-Stachowicz, Anna Janecka, Urszula Krajewska, Marek Mirowski, Tomasz Janecki

**Affiliations:** 1Institute of Organic Chemistry, Lodz University of Technology, Żeromskiego 116, 90-924 Łódź, Poland; tomasz.bartosik@p.lodz.pl (T.B.); jacek.kedzia@p.lodz.pl (J.K.); 2Department of Biomolecular Chemistry, Medical University of Łódź, Mazowiecka 6/8, 92-215 Łódź, Poland; joanna.drogosz@studumed.lodz.pl (J.D.-S.);; 3Department of Pharmaceutical Biochemistry and Molecular Diagnostics, Faculty of Pharmacy, Medical University of Łódź, Muszyńskiego 1, 90-151, Łódź, Poland; ukrajewska@o2.pl (U.K.); marek.mirowski@umed.lodz.pl (M.M.)

**Keywords:** methylidenedihydropyran-4-ones, Michael addition, Horner-Wadsworth-Emmons olefination, cytotoxic activity, apoptosis, cell cycle

## Abstract

In our continuous search for new, relatively simple 2-alkylidene-1-oxoheterocycles as promising anticancer drug candidates, herein we report an efficient synthesis of 2,2,6-trisubstituted 5-methylidenetetrahydropyran-4-ones. These compounds were obtained in a four step reaction sequence, in which starting diethyl 2-oxopropylphosphonate was transformed into 2,2-disubstituted 5-diethoxyphosphoryldihydropyran-4-ones, followed by Michael addition of various Grignard reagents and Horner-Wadsworth-Emmons reaction of the obtained adducts with formaldehyde. Stereochemistry of the intermediate Michael adducts is also discussed. Final 5-methylidenetetrahydropyran-4-ones were tested for their possible antiproliferative effect against three cancer cell lines and 6-isopropyl-2,2-dimethyl-5-methylidenetetrahydropyran-4-one (**11c**), which showed very high cytotoxic activity against HL-60 human leukemia cells and was three times more active than known anticancer drug carboplatin, was selected for further biological evaluation, in order to disclose its mechanism of action. The obtained results indicated that **11c** induced apoptosis in HL-60 cells and caused the arrest of the cell cycle in the G2/M phase, which may suggest its cytotoxic and cytostatic activity.

## 1. Introduction

In the continuation of our search for relatively simple structural motifs with anticancer potential [1,2,3], we turned our attention to 3-alkylidenetetrahydropyran-4-ones **1**. Compounds containing this skeleton are common in nature and display interesting biological activities. This structural motif is present in a number of biologically active and medicinally important natural products. For example, norperovskatone **2**, recently isolated from *Perovskia atriplicifolia*, possesses remarkable anti-HBV (Hepatitis B Virus) activity [4] and (+)-okilactomycin **3 [5]** exhibits in vitro cytotoxicity against a number of human cancer cell lines, including lymphoid leukemia L1210 or leukemia P388. This compound also shows in vivo activity against Ehrlich ascites carcinoma. Chrolactomycin **4** displays significant cytotoxicity in vitro against several human cancer cell lines: ACHN, A431, MCF-7, and T24 [6]. The structures of natural compounds containing 3-alkylidenetetrahydropyran-4-one motif are shown in Figure 1.

However, literature search revealed that synthetic methodologies that can be applied to the preparation of 3-alkylidenetetrahydropyran-4-ones **1** with diverse substitution patterns are limited. There are several methods which enable the introduction of arylidene group onto the plain tetrahydropyran-4-one ring using a reaction with aromatic aldehydes in the presence of catalytic amount of TMSNMe_2_ and MgBr_2_ Et_2_O [7] or pyrrolidine [8]. There are also two literature reports which describe preparation of specific 3-methylidenetetrahydropyran-4-ones, and thus 8-methylidene-9-oxo-6-oxaspiro [5,6] decane was prepared by carbocyclization of unsaturated thioesters under palladium catalysis [9] and 2-butyl-6,6-dimethyl-3-methylidenetetrahydropyran-4-one was obtained in a three-step reaction sequence starting from 6-ethoxy-2-methyl-5-diphenylphosphorylhex-3-yn-5-en-2-ol [10].

In this report, we present a novel, general, and efficient synthesis of 2,2,6-trisubstituted 5-methylidenetetrahydropyran-4-ones **11** using Horner-Wadsworth-Emmons methodology [11,12]. The obtained compounds have been evaluated for their possible cytotoxic activity against three human cancer cell lines: NALM-6 and HL-60 leukemia and the MCF-7 breast cancer cell line.

## 2. Results and Discussion

### 2.1. Chemistry

We start our research from the synthesis of diethyl 4-hydroxy-2-oxoalkylphosphonates **7a**–**d** using a procedure described by Wada and coworkers [13]. In this respect, diethyl 2-oxopropylphosphonate (**5**) was reacted with 1,1 equiv. of sodium hydride at r.t. and next with 1,1 equiv. of 2,5 M *n*-BuLi solution in hexane at −40 °C to obtain dianion **6**, which, after cooling to −70 °C was treated with symmetric ketones, such as acetone, cyclohexanone, benzophenone, and fluoren-9-one (Scheme 1). After standard work-up, expected phosphonates **7a–d** were obtained in moderate to good yields (Table 1). Next, **7a**–**d** were subjected to a reaction with dimethylformamide dimethyl acetal (DMF-DMA) in the presence of trifluoroborate diethyl etherate BF_3_·Et_2_O, applying toluene as a solvent. Initial results, using one equiv. of DMF-DMA and BF_3_·Et_2_O and performing the reaction at r.t. for 24 h were unsatisfactory with 15–20% conversion to the desired pyran-4-ones **9**. To our delight, the optimization of this reaction was fully successful and carrying out the reaction with 3 equiv. of DMF-DMA and 1 equiv. of BF_3_ Et_2_O for 3 h at 100 °C (for pyran-4-ones **9a**–**c**) or at r.t. (for pyran-4-one **9d**) gave desired 3-diethoxyphosphoryldihydropyran-4-ones **9a**–**d** in good yields (Table 1). Apparently, this reaction proceeds via the Knoevenagel type condensation, with the formation of dimethylaminovinylphosphonates **8a**–**d** as intermediates, followed by intramolecular oxy-Michael cyclization. When progress of these reactions was monitored by ^31^P NMR technique, signals with chemical shift ~25 ppm initially appeared, which could be attributed to intermediate vinylphosphonates **8**. Only when BF_3_·Et_2_O was added to the reaction mixture did these signals gradually disappear with the simultaneous appearance of pyran-4-ones **9** signals.

With 3-diethoxyphosphoryldihydropyran-4-ones **9a**–**d** in hand, we used them as Michael acceptors in the reaction with Grignard reagents, to introduce various substituents into position 2. The addition of ethyl-, *n*-butyl- and *i*-propylmagnesium chlorides proceeded smoothly when three equiv. of Grignard reagent were used, while the addition of phenylmagnesium chloride was effective only in the presence of 3 equiv. of copper iodide (I). Standard work-up of the reaction mixtures followed by column chromatography purification furnished 6-substituted 5-diethoxyphosphoryltetrahydropyran-4-ones **10a**–**p** in good yields (Scheme 2, Table 2). Interestingly, pyran-4-ones **10a**–**p** were formed as mixtures of diasteroisomers *trans*-**10a**–**p** and *cis*-**10a**–**p** along with the enol form, enol-**10a**–**p**.

Careful analysis of ^1^H and ^13^C NMR spectra allowed us not only to propose the relative configuration of both diastereoisomers but also their preferred conformation. For example, coupling constants ^3^*J*_H5-H6_, ^3^*J*_H6-P_ and ^3^*J*_C1′-P_ for *trans*-**10f** and *cis*-**10f** (Figure 2) unambiguously indicated diequatorial and axial/equatorial position of *n*-Bu and phosphoryl groups in these diastereoisomers, respectively. Because similar sets of coupling constants were observed for other pairs of diastereoisomers we believe that this assignment is valid for all 6-substituted 5-diethoxyphosphoryltetrahydropyran-4-ones **10a**–**p**. In turn enol form of pyran-4-ones **10a**–**p** had characteristic doublets of hydroxyl group with chemical shift in the range of 11–12 ppm and coupling constant ^4^*J*_H-P_ ~1 Hz. Presence of such coupling constant is yet another confirmation of the existence of resonance assisted hydrogen bond (RAHB) in organophosphorus compounds [14]. Relative ratios of *trans*-, *cis*- and enol-**10a**–**p**, taken from the corresponding ^31^P NMR spectra, are given in Table 2. 

In the final step, the target 2,2,6-triisubstituted-5-methylidenetetrahydropyran-4-ones **11a**–**p** were obtained by olefination of formaldehyde using 5-diethoxyphosphoryltetrahydropyran-4-ones **10a**–**p** as Horner-Wadsworth-Emmons reagents (Scheme 3). Very good yields were obtained when formalin and potassium carbonate were used and the reaction mixture was stirred at 0 °C for 2 h (Table 2). However, to get 2,2-disubstituted 6-isopropyl-5-methylidenetetrahydropyran-4-ones **11c**,**g**,**k**,**o** in reasonable yields (60–70%), stirring of the reaction mixture at r.t. for 3 h was necessary.

### 2.2. Biological Screening of Novel Pyranones

Novel 2,2,6-trisubstituted 5-methylidenetetrahydropyran-4-ones **11a–p** were tested for their possible antiproliferative activity on two human leukemia cell lines, acute promyelocytic leukemia HL-60 and lymphoblastic leukemia NALM-6, and on a solid tumor-derived human breast adenocarcinoma cell line MCF-7, using the tetrazolium derivative reduction assay (MTT) (Table 3). An anticancer drug, carboplatin, and a sesquiterpene lactone, parthenolide, containing the same α, β-unsaturated carbonyl function as analogs **11a**–**p**, were used as reference compounds. Concentration-response analysis was performed to determine drug concentrations required to inhibit the growth of cancer cells by 50% (IC_50_) after 48 h incubation.

All 16 analogs were found to possess antiproliferative activity in the tested cell lines in the µM range. The IC_50_ values below 5 µM were obtained for 4 compounds (**11b**,**c**,**k**,**o**) on HL-60, 8 compounds (**11a**,**b**,**c**,**f**,**g**,**k**,**m**,**o**) on NALM-6, and 1 analog (**11o**) on the MCF-7 cell line. Carboplatin inhibited the growth of all three types of cells with IC_50_ values below 5 µM and the potency order NALM-6 > HL-60 > MCF-7. Parthenolide was about 2-3-fold less cytotoxic than carboplatin on these cell lines. The obtained data indicate that methylidenepyranones **11a**–**p** were more cytotoxic against leukemia cells than against breast cancer cells. Analog **11c** was the most cytotoxic, with a three-fold lower IC_50_ than carboplatin on HL-60 and NALM-6 cells (1.02 and 0.27 µM as compared to 2.9 and 0.7 µM for carboplatin, respectively). None of the analogs were more cytotoxic than carboplatin on MCF-7 cells. 

Analysis of the structure-activity relationship revealed that the cytotoxicity of 5-methylidenetetrahydropyran-4-ones **11a**–**p** strongly depended on the nature of the substituent in position 6 (R^2^) and, to a much lesser extent, on the structure of geminal substituents in position 2 (R^1^, R^1^). This observation can be rationalized with the reasonable assumption that the conjugated methylidene group in pyranones **11** acts as an effective Michael acceptor towards various bionucleophiles through a reaction with mercapto groups (-SH) of cysteine residues in proteins and intracellular glutathione [15]. Such alkylation of cellular thiols disrupts key biological processes and affects multiple targets in cancer cells [16]. Therefore, R^2^ substituents that are much closer to the reaction site (methylidene group) than R^1^ substituents can influence the effectiveness of the alkylation to a much greater extent. 

Among position 2 substituents (R^1^, R^1^), two phenyl groups (**11i**-**l**) or a fluorenyl moiety (**11m**-**p**) were more advantageous for growth inhibition on all three tested cell lines than the non-aromatic substituents. Among analogs with various R^2^ alkyl groups introduced into position 6, those having isopropyl were the most active. The only exception was pyran-4-one **11k** (R^1^ = Ph, R^2^ = *i*-Pr, IC_50_ = 11.9 μM), which was less active than pyran-4-one **11j** (R^1^ = Ph, R^2^ = *n*-Bu, IC_50_ = 6.9 μM) against the MCF-7 cell line. As opposed to position 2, phenyl moiety in position 6 (as R^2^) was the one contributing less to the compounds’ cytotoxicity.

Evidently, the most cytotoxic compound of the series was pyran-4-one **11c** (R^1^ = Me, R^2^ = *i*-Pr), which had the highest activity among all tested analogs on HL-60 and NALM-6 cell lines (IC_50_ = 1.02 μM and 0.27 μM, respectively). Therefore, analog **11c** was selected for further evaluation of its anticancer activity. Since the mortality rates for acute myeloid leukemia are very high [17], necessitating the search for novel chemotherapeutic candidates, additional tests were performed on the HL-60 cell line. 

We wanted to determine whether the effect of **11c** on cell viability was due to apoptosis. DNA damage and the subsequent induction of apoptosis is a primary cytotoxic mechanism of many anticancer agents. 

Cytotoxicity of **11c** in HL-60 cells was once again measured after a shorter incubation time (24 h) using a broad range of analog concentrations (0–25 µM) and the obtained IC_50_ value was 2.2 µM (Figure 3A). Phosphorylation of H2AX histone was used as a marker of DNA double-strand breaks. To assess the ability of **11c** to phosphorylate H2AX, HL-60 cells were incubated with this compound for 24 h at 2.2 and 4.4 µM concentration (IC_50_ and 2 IC_50_, respectively). At 4.4 µM, **11c** caused a significant H2AX phosphorylation, indicative of DNA damage (Figure 3B). The induction of apoptosis was assessed by detection of the 89 kDa-cleaved fragment of PARP (poly (ADP-ribose) polymerase). The tested analog caused about 4-fold increase in the number of cleaved PARP positive (apoptotic) cells as compared to the control (Figure 3C).

Analysis of the cell cycle phases was then performed using flow cytometry. After 24 h incubation with **11c**, HL-60 cells were treated with a fluorescent dye (4,6-diamidino-2-phenylindole, DAPI) that quantitatively stains DNA. The fluorescence intensity of the stained cells correlates with the amount of DNA they contain and identifies cell cycle position in the major phases (G0/G1 *versus* S *versus* G2/M phase). Representative profiles are shown in Figure 4. Analog **11c** at a concentration of 4.4 µM caused a significant increase of cells in the G2/M phase (from 18.9% in control to 31.4%), suggesting an antimitotic effect. Together, the obtained data may indicate that the novel pyranone **11c** produced both cytotoxic and cytostatic effects.

## 3. Materials and Methods

### 3.1. Chemistry

#### 3.1.1. General Information

NMR spectra were recorded on a Bruker DPX 250 (Billerica, MA, USA) or Bruker Avance II instrument (Billerica, MA, USA) at 250.13 MHz or 700 MHz for ^1^H, 62.9 MHz or 176 MHz for ^13^C, and 101.3 MHz for ^31^P NMR with tetramethylsilane used as an internal and 85% H_3_PO_4_ as an external standard. ^31^P NMR spectra were recorded using broadband proton decoupling. IR spectra were recorded on a Bruker Alpha ATR spectrophotometer. Melting points were determined in open capillaries and are uncorrected. Column chromatography was performed on silica gel 60 (230–400 mesh) (Aldrich, Saint Louis, MO, US). Thin-layer chromatography was performed on the pre-coated TLC sheets of silica gel 60 F254 (Aldrich). The purity of the synthesized compounds was confirmed by combustion elemental analyses (CH, elemental analyzer EuroVector 3018, Elementar Analysensysteme GmbH, Langenselbold, Germany. MS spectra of intermediates were recorded on Waters 2695-Waters ZQ 2000 LC/MS apparatus. EI mass spectra of the final compounds were recorded on a GCMS-QP2010 ULTR A instrument (Shimadzu, Kyoto, Japan). Mass spectra were obtained using the following operating conditions: electron energy of 70 eV and ion source temperature of 200 °C. Samples were introduced via a direct insertion probe heated from 30 °C to 300 °C. All reagents and starting materials were purchased from commercial vendors and used without further purification. Organic solvents were dried and distilled prior to use. Standard syringe techniques were used for transferring dry solvents.

General procedures and characterization data for diethyl 4-hydroxy-2-oxoalkylphosphonates **7a**–**d**, 3-diethoxyphosphoryldihydropyran-4-ones **9a**–**d** and 2,2-dialkyl(diaryl)-6-alkyl(aryl)-5-methylenetetrahydro -4*H*-pyran-4-ones **10a**–**p** are given in Appendix A.

#### 3.1.2. General Procedure for the Synthesis of 2,2,6-Triisubstituted-5-Methylidenetetrahydropyran-4-Ones **11a–p**

To a vigorously stirred solution of 5-diethoxyphosphoryltetrahydropyran-4-ones **10a**–**p** (0.2 mmol) in dry THF (2 mL), formaldehyde (36–38% solution in water, 0.167 mL, ca. 2.00 mmol) was added at 0 °C, followed by addition of K_2_CO_3_ (55.3 mg, 0.40 mmol) in water (0.56 mL). The resulting mixture was stirred vigorously at 0 °C for 2 h (for pyran-4-ones **10a**,**b**,**d**–**f**,**h**–**j**,**l**–**n**,**p**) or at r.t. for 3 h (for pyran-4-ones **10c**,**g**,**k**,**o**). Next Et_2_O (10 mL) was added and the layers were separated. The water fraction was washed with Et_2_O (5 mL). Organic fractions were combined, washed with brine (10 mL), and dried over MgSO_4_. The solvents were evaporated under reduced pressure and the resulting crude product was purified by column chromatography (eluent DCM).

*6-Ethyl-2,2-dimethyl-5-methylidenetetrahydro-4H-pyran-4-one* (**11a**) (32.6 mg, 97%) Colorless oil. ^1^H NMR (700 MHz, Chloroform-*d*) δ 0.97 (t, *J* = 7.3 Hz, 3H), 1.28 (s, 3H), 1.29 (s, 3H), 1.60–1.69 (m, 1H), 1.79–1.89 (m, 1H), 2.39–2.51 (m, 2H), 4.47 (ddd, *J* = 5.6, 2.0, 2.0 Hz, 1H), 5.27 (dd, *J* = 2.0, 1.0 Hz, 1H), 6.14 (dd, *J* = 2.0, 1.0 Hz, 1H). ^13^C NMR (176 MHz, Chloroform-*d*) δ 9.3, 25.3, 28.0, 30.8, 51.1, 72.3, 72.9, 119.6, 144.9, 197.9. EI-MS [M]^+^ = 168.0. Anal. Calcd. for C_10_H_16_O_2_: C, 71.39; H, 9.59. Found: C, 71.21; H, 9.61.

*6-Butyl-2,2-dimethyl-5-methylidenetetrahydro-4H-pyran-4-one* (**11b**) (37.7 mg, 96%) Colorless oil. ^1^H NMR (700 MHz, Chloroform-*d*) δ 0.91 (t, *J* = 7.2 Hz, 3H), 1.27 (s, 3H), 1.28 (s, 3H), 1.30–1.42 (m, 3H), 1.43–1.52 (m, 1H), 1.62 (dddd, *J* = 14.1, 10.2, 7.8, 4.7 Hz, 1H), 1.77 (dddd, *J* = 14.1, 10.6, 5.4, 3.7 Hz, 1H), 2.46 (d, *J* = 16.7 Hz, 2H), 4.50 (dddd, *J* = 7.8, 3.9, 2.0, 2.0 Hz, 1H), 5.27 (dd, *J* = 2.0, 1.0 Hz, 1H), 6.11 (dd, *J* = 2.1, 1.0 Hz, 1H). ^13^C NMR (176 MHz, Chloroform-*d*) δ 14.2, 22.8, 25.4, 27.3, 30.8, 34.7, 51.1, 71.8, 72.4, 119.4, 145.3, 197.9. EI-MS [M]^+^ = 196.0. Anal. Calcd. for C_12_H_20_O_2_: C, 73.43; H, 10.27. Found: C, 73.60; H, 10.30.

*6-Isopropyl-2,2-dimethyl-5-methylidenetetrahydro-4H-pyran-4-one* (**11c**) (18.6 mg, 51%) Colorless oil. ^1^H NMR (700 MHz, Chloroform-*d*) δ 0.86 (d, *J* = 6.8 Hz, 3H), 1.01 (d, *J* = 6.8 Hz, 3H), 1.26 (s, 3H), 1.29 (s, 3H), 1.96 (heptd, *J* = 6.8, 3.2 Hz, 1H), 2.37–2.48 (m, 2H), 4.44 (ddd, *J* = 3.2, 2.0, 2.0 Hz, 1H), 5.27 (dd, *J* = 2.0, 1.1 Hz, 1H), 6.20 (dd, *J* = 2.0, 1.1 Hz, 1H). ^13^C NMR (176 MHz, Chloroform-*d*) δ 15.7, 19.4, 25.0, 30.8, 33.7, 51.0, 71.8, 76.8, 120.6, 144.1, 198.1. EI-MS [M]^+^ = 182.0. Anal. Calcd. for C_11_H_18_O_2_: C, 72.49; H, 9.95. Found: C, 72.71; H, 9.92.

*2,2-Dimethyl-5-methylidene-6-phenyltetrahydro-4H-pyran-4-one* (**11d**) (32.9 mg, 76%) Colorless oil. ^1^H NMR (700 MHz, Chloroform-*d*) δ 1.40 (s, 3H), 1.42 (s, 3H), 2.65 (s, 2H), 4.82 (dd, *J* = 2.2, 1.2 Hz, 1H), 5.52 (dd, *J* = 2.2, 2.2 Hz, 1H), 6.15 (dd, *J* = 2.2, 1.2 Hz, 1H), 7.30 – 7.34 (m, 1H), 7.35 – 7.40 (m, 4H). ^13^C NMR (176 MHz, Chloroform-*d*) δ 25.5, 30.8, 51.5, 73.5, 76.0, 123.0, 128.0 (2 × C), 128.4, 128.7 (2 × C), 140.5, 145.4, 197.1. EI-MS [M]^+^ = 216.0. Anal. Calcd. for C_14_H_16_O_2_: C, 77.75; H, 7.46. Found: C, 77.71; H, 7.47.

*2-Ethyl-3-methylidene-1-oxaspiro [5.5]undecan-4-one (**11e**)* (41.2 mg, 99%) Colorless oil. ^1^H NMR (700 MHz, Chloroform-*d*) δ 1.05 (t, *J* = 7.3 Hz, 3H), 1.25–1.92 (m, 12H), 2.38 (d, *J* = 16.7 Hz, 1H), 2.44 (d, *J* = 16.7 Hz, 1H), 4.37 (dddd, *J* = 7.8, 3.6, 2.1, 2.1 Hz, 1H), 5.25 (dd, *J* = 2.1, 1.0 Hz, 1H), 6.11 (dd, *J* = 2.1, 1.0 Hz, 1H). ^13^C NMR (176 MHz, Chloroform-*d*) δ 9.9, 21.8, 21.8, 25.5, 28.1, 33.2, 39.3, 50.9, 71.7, 73.1, 119.3, 145.6, 198.2. EI-MS [M]^+^ = 208.0. Anal. Calcd. for C_13_H_20_O_2_: C, 74.96; H, 9.68. Found: C, 74.76; H, 9.70.

*2-Butyl-3-methylidene-1-oxaspiro[5.5]undecan-4-one (**11f**)* (43.5 mg, 92%) Colorless oil. ^1^H NMR (700 MHz, Chloroform-*d*) δ 0.92 (t, *J* = 7.2 Hz, 3H), 1.18–1.91 (m, 16H), 2.37 (d, *J* = 16.7 Hz, 1H), 2.43 (d, *J* = 16.7 Hz, 1H), 4.32–4.49 (m, 1H), 5.25 (dd, *J* = 2.1, 1.0 Hz, 1H), 6.08 (dd, *J* = 2.1, 1.0 Hz, 1H). ^13^C NMR (176 MHz, Chloroform-*d*) δ 14.2, 21.8, 21.8, 22.8, 25.5, 27.7, 33.2, 34.8, 39.3, 50.9, 70.6, 73.1, 119.1, 145.9, 198.2. EI-MS [M]^+^ = 236.0. Anal. Calcd. for C_15_H_24_O_2_: C, 76.23; H, 10.24. Found: C, 76.18; H, 10.25.

*2-Isopropyl-3-methylidene-1-oxaspiro[5.5]undecan-4-one* (**11g**) (24.5 mg, 55%) Colorless oil. ^1^H NMR (700 MHz, Chloroform-*d*) δ 0.86 (d, *J* = 6.8 Hz, 3H), 1.07 (d, *J* = 6.8 Hz, 3H), 1.16–1.91 (m, 10H), 2.00 (hd, *J* = 6.8, 3.0 Hz, 1H), 2.32 (d, *J* = 17.1 Hz, 1H), 2.38 (d, *J* = 17.1 Hz, 1H), 4.37 (ddd, *J* = 3.0, 2.2, 2.2 Hz, 1H), 5.24 (dd, *J* = 2.2, 1.1 Hz, 1H), 6.18 (dd, *J* = 2.2, 1.1 Hz, 1H). ^13^C NMR (176 MHz, Chloroform-*d*) δ 15.3, 19.7, 21.7, 21.9, 25.6, 32.9, 33.4, 39.2, 50.8, 72.4, 75.2, 120.1, 144.7, 198.3. EI-MS [M]^+^ = 222.0. Anal. Calcd. for C_14_H_22_O_2_: C, 75.63; H, 9.97. Found: C, 75.80; H, 9.96.

*3-Methylidene-2-phenyl-1-oxaspiro[5.5]undecan-4-one* (**11h**) (38.5 mg, 75%) White solid mp 128 – 130 °C. ^1^H NMR (700 MHz, Chloroform-*d*) δ 1.04–2.12 (m, 10H), 2.57 (d, *J* = 16.5 Hz, 1H), 2.64 (d, *J* = 16.5 Hz, 1H), 4.82 (dd, *J* = 2.2, 1.2 Hz, 1H), 5.48 (dd, *J* = 2.2, 2.2 Hz, 1H), 6.14 (dd, *J* = 2.2, 1.2 Hz, 1H), 7.31–7.35 (m, 1H), 7.36–7.40 (m, 4H). ^13^C NMR (176 MHz, Chloroform-*d*) δ 21.8, 21.9, 25.5, 33.7, 39.2, 50.6, 74.4, 74.9, 122.7, 128.0 (2 × C), 128.2, 128.6 (2 × C), 140.8, 145.8, 197.4. EI-MS [M]^+^ = 256.0. Anal. Calcd. for C_17_H_20_O_2_: C, 79.65; H, 7.86. Found: C, 79.87; H, 7.85.

*6-Ethyl-5-methylidene-2,2-diphenyltetrahydro-4H-pyran-4-one* (**11i**) (55.6 mg, 95%) Pale yellow oil. ^1^H NMR (700 MHz, Chloroform-*d*) δ 1.14 (t, *J* = 7.3 Hz, 3H), 1.79–1.87 (m, 1H), 1.89–1.98 (m, 1H), 2.90 (d, *J* = 17.1 Hz, 1H), 3.58 (d, *J* = 17.2 Hz, 1H), 4.21 (dddd, *J* = 7.8, 3.9, 2.2, 2.2 Hz, 1H), 5.19 (dd, *J* = 2.2, 1.0 Hz, 1H), 6.13 (dd, *J* = 2.2, 1.0 Hz, 1H), 7.26 (m, 10H). ^13^C NMR (176 MHz, Chloroform-*d*) δ 9.7, 28.0, 50.98, 73.5, 80.1, 120.2, 125.1 (2 × C), 127.0, 127.9 (2 × C), 127.9, 128.3 (2 × C), 128.8 (2 × C), 142.2, 144.4, 147.6, 196.5. EI-MS [M]^+^ = 292.0. Anal. Calcd. for C_20_H_20_O_2_: C, 82.16; H, 6.90. Found: C, 82.37; H, 6.89.

*6-Butyl-5-methylidene-2,2-diphenyltetrahydro-4H-pyran-4-one* (**11j**) (55.8 mg, 91%) White solid 141 – 143 °C. ^1^H NMR (700 MHz, Chloroform-*d*) δ 0.88–0.98 (m, 1H), 1.02 (t, *J* = 7.3 Hz, 3H), 1.42–1.48 (m, 1H), 1.52–1.61 (m, 1H), 1.71–1.79 (m, 1H), 1.85–1.92 (m, 2H), 2.94 (d, *J* = 17.1 Hz, 1H), 3.63 (d, *J* = 17.1 Hz, 1H), 4.28 (dddd, *J* = 6.9, 4.4, 2.1, 2.1 Hz, 1H), 5.25 (dd, *J* = 2.1, 1.1 Hz, 1H), 6.16 (dd, *J* = 2.1, 1.1 Hz, 1H), 7.26 (m, 10H). ^13^C NMR (176 MHz, Chloroform-*d*) δ 14.2, 22.8, 27.4, 34.7, 51.0, 72.4, 80.1, 120.1, 125.1 (2 × C), 127.0, 127.9 (2 × C), 127.9, 128.3 (2 × C), 128.8 (2 × C), 142.2, 144.8, 147.6, 196.6. EI-MS [M]^+^ = 320.0. Anal. Calcd. for C_22_H_24_O_2_: C, 82.46; H, 7.55. Found: C, 82.66; H, 7.53.

*6-Isopropyl-5-methylidene-2,2-diphenyltetrahydro-4H-pyran-4-one* (**11k**) (34.9 mg, 57%) Pale yellow oil. ^1^H NMR (700 MHz, Chloroform-*d*) δ 1.09 (d, *J* = 6.8 Hz, 3H), 1.23 (d, *J* = 6.9 Hz, 3H), 2.10 (heptd, *J* = 6.8, 2.8 Hz, 1H), 2.90 (d, *J* = 17.5 Hz, 1H), 3.59 (d, *J* = 17.5 Hz, 1H), 4.16–4.29 (m, 1H), 5.17 (dd, *J* = 2.0, 1.0 Hz, 1H), 6.21 (dd, *J* = 2.0, 1.0 Hz, 1H), 7.18–7.51 (m, 10H). ^13^C NMR (176 MHz, Chloroform-*d*) δ 15.7, 19.9, 33.5, 50.8, 77.0, 79.6, 121.2, 125.0 (2 × C), 127.0, 127.89 (3 × C), 128.3 (2 × C), 128.8 (2 × C), 142.0, 143.3, 147.7, 196.5. EI-MS [M]^+^ = 306.0. Anal. Calcd. for C_21_H_22_O_2_: C, 82.32; H, 7.24. Found: C, 82.07; H, 7.22.

*5-Methylidene-2,2,6-triphenyltetrahydro-4H-pyran-4-one* (**11l**) (57.9 mg, 85%) Pale yellow oil. ^1^H NMR (700 MHz, Chloroform-*d*) δ 3.12 (d, *J* = 17.0 Hz, 1H), 3.70 (d, *J* = 17.0 Hz, 1H), 4.76 (dd, *J* = 2.2, 1.1 Hz, 1H), 5.20 (dd, *J* = 2.2, 2.2 Hz, 1H), 6.13 (dd, *J* = 2.2, 1.1 Hz, 1H), 7.17–7.46 (m, 15H). ^13^C NMR (176 MHz, Chloroform-*d*) δ 51.5, 76.6, 81.3, 123.3, 125.2 (2 × C), 127.2, 128.0 (2 × C), 128.2 (3 × C), 128.4 (2 × C), 128.5, 128.7 (2 × C), 129.0 (2 × C), 140.3, 142.0, 145.2, 147.2, 196.0. EI-MS [M]^+^ = 340.0. Anal. Calcd. for C_24_H_20_O_2_: C, 84.68; H, 5.92. Found: C, 84.45; H, 5.90.

*6′-Ethyl-5′-methylidene-5′,6′-dihydrospiro[fluorene-9,2′-pyran]-4′(3′H)-one* (**11m**) (47.6 mg, 82%) Pale yellow oil. ^1^H NMR (700 MHz, Chloroform-*d*) δ 0.99 (t, *J* = 7.4 Hz, 3H), 1.78–1.88 (m, 1H), 1.96 (ddt, *J* = 14.6, 7.3, 3.6 Hz, 1H), 2.87 (d, *J* = 16.7 Hz, 1H), 3.12 (d, *J* = 16.7 Hz, 1H), 5.00 (dddd, *J* = 6.1, 4.0, 2.1, 2.1 Hz, 1H), 5.53 (dd, *J* = 2.1, 1.2 Hz, 1H), 6.49 (dd, *J* = 2.1, 1.2 Hz, 1H), 7.24 (dd, *J* = 8.4, 7.5 Hz, 1H), 7.33 (dd, *J* = 8.4, 7.3 Hz, 1H), 7.39 (dd, *J* = 8.4, 7.6 Hz, 2H), 7.41 (d, *J* = 7.5 Hz, 1H), 7.46 (d, *J* = 7.4 Hz, 1H), 7.64 (d, *J* = 7.5 Hz, 1H), 7.68 (d, *J* = 7.6 Hz, 1H). ^13^C NMR (176 MHz, Chloroform-*d*) δ 9.0, 27.8, 47.3, 75.5, 82.6, 120.1, 120.8, 121.0, 123.9, 124.7, 127.6, 128.5, 129.5, 129.5, 139.0, 140.3, 144.7, 145.3, 147.4, 196.3. EI-MS [M]^+^ = 290.0. Anal. Calcd. for C_20_H_18_O_2_: C, 82.73; H, 6.25. Found: C, 83.01; H, 6.23.

*6′-Butyl-5′-methylidene-5′,6′-dihydrospiro[fluorene-9,2′-pyran]-4′(3′H)-one* (**11n**) (63.04 mg, 99%) Pale yellow oil. ^1^H NMR (700 MHz, Chloroform-*d*) δ 0.87 (t, *J* = 7.3 Hz, 3H), 1.26–1.49 (m, 4H), 1.79 (dddd, *J* = 14.0, 10.5, 7.1, 4.8 Hz, 1H), 1.90 (dddd, *J* = 14.3, 10.7, 5.7, 3.8 Hz, 1H), 2.88 (d, *J* = 16.6 Hz, 1H), 3.09 (d, *J* = 16.7 Hz, 1H), 5.03 (dddd, *J* = 7.2, 4.0, 2.1, 2.1 Hz, 1H), 5.53 (dd, *J* = 2.1, 1.2 Hz, 1H), 6.46 (dd, *J* = 2.1, 1.2 Hz, 1H), 7.24 (dd, *J* = 8.6, 7.5 Hz, 1H), 7.33 (dd, *J* = 8.6, 7.4 Hz, 1H), 7.37–7.40 (m, 2H), 7.41 (d, *J* = 7.5 Hz, 1H), 7.44 (d, *J* = 7.5 Hz, 1H), 7.63 (d, *J* = 7.6 Hz, 1H), 7.68 (d, *J* = 7.5 Hz, 1H). ^13^C NMR (176 MHz, Chloroform-*d*) δ 14.1, 22.8, 26.9, 34.6, 47.3, 74.7, 82.7, 120.1, 120.8, 120.8, 124.0, 124.7, 127.7, 128.5, 129.5, 129.5, 139.0, 140.3, 145.22, 145.4, 147.5, 196.4. EI-MS [M]^+^ = 318.0. Anal. Calcd. for C_22_H_22_O_2_: C, 82.99; H, 6.96. Found: C, 82.75; H, 6.97.

*6′-Isopropyl-5′-methylidene-5′,6′-dihydrospiro[fluorene-9,2′-pyran]-4′(3′H)-one* (**11o**) (35.3 mg, 58%) Pale yellow oil. ^1^H NMR (700 MHz, Chloroform-*d*) δ 0.99 (d, *J* = 6.9 Hz, 3H), 1.01 (d, *J* = 6.8 Hz, 3H), 2.11–2.17 (m, 1H), 2.71 (d, *J* = 17.1 Hz, 1H), 3.15 (d, *J* = 17.1 Hz, 1H), 4.86 (ddd, *J* = 3.2, 2.1 Hz, 1H), 5.55 (dd, *J* = 2.1, 1.2 Hz, 1H), 6.56 (dd, *J* = 2.2, 1.2 Hz, 1H), 7.21 (dd, *J* = 8.6, 7.5 Hz, 1H), 7.34 (dd, *J* = 8.5, 7.6 Hz, 1H), 7.36–7.43 (m, 3H), 7.47 (d, *J* = 7.5 Hz, 1H), 7.63 (d, *J* = 7.5 Hz, 1H), 7.68 (d, *J* = 7.5 Hz, 1H). ^13^C NMR (176 MHz, Chloroform-*d*) δ 15.7, 19.6, 33.9, 47.7, 79.3, 82.4, 120.0, 120.8, 122.0, 124.0, 124.9, 127.7, 128.5, 129.4, 129.5, 139.0, 140.4, 144.3, 145.2, 147.6, 196.3. EI-MS [M]^+^ = 304.0. Anal. Calcd. for C_21_H_20_O_2_: C, 82.86; H, 6.62. Found: C, 82.92; H, 6.61.

*5′-Methylidene-6′-phenyl-5′,6′-dihydrospiro[fluorene-9,2′-pyran]-4′(3′H)-one* (**11p**) (66.3 mg, 98%) Pale yellow oil. ^1^H NMR (700 MHz, Chloroform-*d*) δ 3.02 (d, *J* = 16.6 Hz, 1H), 3.30 (d, *J* = 16.5 Hz, 1H), 5.05 (dd, *J* = 2.1, 1.4 Hz, 1H), 5.99 (dd, *J* = 2.1, 2.1 Hz, 1H), 6.47 (dd, *J* = 2.1, 1.4 Hz, 1H), 7.28–7.44 (m, 9H), 7.51 (d, *J* = 7.6 Hz, 1H), 7.53 (d, *J* = 7.4 Hz, 1H), 7.63 (d, *J* = 7.5 Hz, 1H), 7.69 (d, *J* = 7.6 Hz, 1H). ^13^C NMR (176 MHz, Chloroform-*d*) δ 47.6, 78.6, 83.4, 120.2, 120.9, 124.1, 124.5, 124.8, 127.8, 128.2 (2 × C), 128.6 (2 × C), 128.7 (2 × C), 129.7, 129.7, 139.2, 140.0, 140.5, 144.8, 145.71, 146.9, 195.8. EI-MS [M]^+^ = 338.0. Anal. Calcd. for C_24_H_18_O_2_: C, 85.18; H, 5.36. Found: C, 85.01; H, 5.35.

### 3.2. Biology

#### 3.2.1. Cell Culture

The breast cancer MCF-7, leukemia HL-60, and NALM-6 cell lines were obtained from the European Collection of Cell Cultures (ECACC). The MCF-7 cells were cultured in Minimum Essential Medium Eagle (MEME, Sigma-Aldrich, St. Louis, MO, USA) with glutamine (2 mM), Men Non-essential amino acid solution, gentamycin (5 μg/mL) and 10% heat-inactivated fetal bovine serum (FBS). HL-60 and NALM-6 cells were maintained in RPMI 1640 + Glutamax medium (Gibco/Life Technologies, Carlsbad, CA, USA), supplemented with 10% FBS, streptomycin and penicillin. Cells were maintained in the logarithmic phase at 37 °C under a 5% carbon dioxide atmosphere.

#### 3.2.2. Cytotoxicity Determination (MTT Assay)

The MTT assay was performed according to a known procedure [18]. The cells were incubated with the analogs for 48 or 24 h. The absorbance of the blue formazan product was measured at 560 nm using FlexStation 3 Multi-Mode Microplate Reader (Molecular Devices, LLC, San Jose, CA, USA) and compared with the control (untreated cells). All experiments were performed in triplicate.

#### 3.2.3. Analysis of DNA Damage and Apoptosis by Flow Cytometry

DNA damage and apoptotic cell death were determined by flow cytometry using fluorescent antibodies for phosphorylated H2AX (Alexa Fluor 647 Mouse Anti-H2AX (pS139)) and cleaved PARP (PE Mouse Anti-Cleaved PARP (Asp214) Antibody) (BD Bioscience, San Jose, CA, USA), according to the manufacturer’s guidelines, as described in detail elsewhere [19].

#### 3.2.4. Cell Cycle Analyses by Flow Cytometry

The cell cycle kinetics were determined using DAPI staining solution (BD Bioscience), according to the manufacturer guidelines, as described in detail elsewhere [20]. Cells were analyzed by flow cytometry using CytoFLEX (Beckman Coulter, Inc., Brea, CA, USA).

#### 3.2.5. Statistical Analysis

Statistical analyses were performed using Prism 4.0 (GraphPad Software Inc., San Diego, CA, USA). The data were expressed as means ± SEM. Statistical comparisons were assessed by a one-way ANOVA, followed by a post-hoc multiple comparison Student-Newman-Keuls test or Student t test. A probability level of 0.05 or lower was considered statistically significant; *** *p* < 0.001, ** *p* < 0.01, * *p* < 0.05.

## 4. Conclusions

The reported results showed that biologically important 5-methylidenetetrahydropyran-4-ones **11** can be effectively synthesized using the Horner-Wadsworth-Emmons reaction of easily accessible 3-diethoxyphosphoryldihydropyran-4-ones **10** with formaldehyde. Various substituents in position 2 can be introduced to the target compounds performing the reaction of dianion **6** with both, acyclic or cyclic symmetrical ketones. Furthermore, additional substituents can be placed in position **6** by the addition of selected Grignard reagents to 3-diethoxyphosphoryldihydropyran-4-ones **9**. Therefore, the described methodology has a broad scope, as a variety of substituents can be introduced in these positions. Currently, the reaction of aldehydes and unsymmetrical ketones with dianion **6** is investigated to broaden the scope of the presented methodology. The biological evaluation of the obtained 5-methylidenetetrahydropyran-4-ones **11** confirmed our assumption that these compounds possess a strong cytotoxic activity against several cancer cell lines. Investigation of structure-activity relationship revealed that the *i*-propyl substituent in position 6 is crucial for activity. 6-Isopropyl-2,2-dimethyl-5-methylidenetetrahydropyran-4-one **11c**, which was recognized as the most active compound against the HL-60 cancer cell line, was further tested in order to disclose its mechanism of action. The preliminary tests indicated that **11c** induced apoptosis in HL-60 cells and caused the arrest of the cell cycle in the G2/M phase, which may suggest its cytotoxic and cytostatic activity. Further biological experiments are necessary to fully determine the mode of action of this analog in leukemic cells.

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
