# Peer review of "Synthesis of 2,2,6-Trisubstituted 5-Methylidene-tetrahydropyran-4-ones with Anticancer Activity†"

_molecules, 2020, doi:10.3390/molecules25030611_

Round 1

Reviewer 1 Report

Introduction is well structured, results are well explained and discussed and conclusions are based on the obtained results.

Despite that, there are some minor changes required for publication.

Figure 2: the compound number (10c) does not match with the one in the table (table 2).

Nomenclature of 11a-11d should be checked according to the general nomenclature that appears in line 206, as they do not match. A general nomenclature revision is required for publication. 

Author Response

Answers to Rewiever 1:

1) Figure 2: the compound number (10c) does not match with the one in the table (table 2).

Answer

Compound number was corrected to 10f

2) Nomenclature of 11a-11d should be checked according to the general nomenclature that appears in line 206, as they do not match. A general nomenclature revision is required for publication.

Answer

A general nomenclature revision for publication has been done. Now all compounds 11 are named as 2,2,6-trisubstituted-5-metylidenepyran-4-ones, what is in accordance with IUPAC nomenclature. Therefore, also the title of the manuscript has been changed from : ”Synthesis of 2,6,6-trisubstituted 3-methylidenetetrahydropyran-4-ones” with anticancer activity to “Synthesis of 2,2,6-trisubstituted 5-methylidenetetrahydropyran-4-ones with anticancer activity”.

Reviewer 2 Report

T. Janecki and co-workers synthesized some new pyran-4-one based compounds to evaluate their antiproliferative properties. A very interesting stereochemical/conformational analysis using the 31P-NMR technique was done. Significative IC50 values of cytotoxicity were found for some synthesized products. Mechanism of action was investigated by flow cytometry studies. The manuscript seems to be well written. A detailed Supp Info File including spectra for all the products is provided. Thus, I recommend publication of this paper after some minor corrections, as follows:

The use of adjectives should be minimum. Novel, efficient, simple, promising, common, important, strong, and other similar words can be found in all the manuscript. In this context, I respectfully recommend modifying the title by “synthesis of 2,6,6-trisubstituted 3-3 methylidenetetrahydropyran-4-ones with anticancer activity” Pg 2 – Ln 65: n should be in italics (n-BuLi); also, in Scheme 1 BF3 x Et2O should be BF3 ∙ Et2O in Scheme 1 iPr should be i-Pr for R2 in Table 2 and also in Table 3 Eq. should be Equiv. throughout the manuscript The SAR analysis was done in a nominal three-four sentences (Pg 5 – Ln 139-147), which needs to be increased substantially. Authors have in hands 16 tested products (with a variety of substituents in C-2 and C-6 positions) and IC50 values for 3 carcinoma cell lines Plots from Fig 3 and Fig 4 should be replaced to improve the sharpness For the 13C-NMR peaks, please correct them by using only one digit after dots It would have been great to complete this interesting work with related computational studies, for instance, docking, QSAR and/or pharmacophore model (e.g. taking 11c as starting point)

Author Response

Answers to Reviewer 2:

The use of adjectives should be minimum. Novel, efficient, simple, promising, common, important, strong, and other similar words can be found in all the manuscript. In this context, I respectfully recommend modifying the title by “synthesis of 2,6,6-trisubstituted 3-3 methylidenetetrahydropyran-4-ones with anticancer activity”

Answer

The title has been changed as recommended and several adjectives have been omitted throughout the manuscript.

Pg 2 – Ln 65: n should be in italics (n-BuLi); also, in Scheme 1 BF3 x Et2O should be BF3 ∙ Et2O in Scheme 1 iPr should be i-Pr for R2 in Table 2 and also in Table 3 Eq. should be Equiv. throughout the manuscript

Answer

Corrected

The SAR analysis was done in a nominal three-four sentences (Pg 5 – Ln 139-147), which needs to be increased substantially. Authors have in hands 16 tested products (with a variety of substituents in C-2 and C-6 positions) and IC50 values for 3 carcinoma cell lines.

Answer

SAR analysis has been expanded (lines 139-179).

Plots from Fig 3 and Fig 4 should be replaced to improve the sharpness

Answer

Fig. 3 and Fig.4 have been provided in better quality.

For the 13C-NMR peaks, please correct them by using only one digit after dots.

Answer

Corrected.

It would have been great to complete this interesting work with related computational studies, for instance, docking, QSAR and/or pharmacophore model (e.g. taking 11c as starting point)

Answer

Further research on 11c is being planned.

Reviewer 3 Report

Molecule-707687 disclosed a series of  3-methylidenetetrahydropyran-4-ones, which showed cytotoxic activity against cancer cell lines HL-60, NALM-6, and MCF-7. Later, the mechanism of the selected compound 11c was further explored. Thus, this paper is recommended to be accepted after a minor revise.

1) The anti-cancer activities of these compounds may result from the high electrophilicity of the Michael acceptor in the structure. Thus, their cytotoxic activity against normal (non-cancer) cells should also be tested.

2) How do you think about the metabolic stability of these compounds in vivo, when considering the existence of Michael acceptor?

3) It seems that R2 showed significant effects on their activities, rather than R1. Could you try to give a possible explanation for this observation? 

4) There are some format errors in Table 3, in which certain lines are missing and "Me, Me" shouldn't be bold.

Author Response

Answers to Reviewer 3

1) The anti-cancer activities of these compounds may result from the high electrophilicity of the Michael acceptor in the structure. Thus, their cytotoxic activity against normal (non-cancer) cells should also be tested.

Answer

Further research on 11c is being planned. It will include testing on the healthy cell line.

2) How do you think about the metabolic stability of these compounds in vivo, when considering the existence of Michael acceptor?

Answer

The compounds are stable in vitro (their stability after 2 months in the refrigerator was assessed by NMR). The metabolic stability in vivo was not tested.

3) It seems that R2 showed significant effects on their activities, rather than R1. Could you try to give a possible explanation for this observation? 

Answer

The paragraph has been added on page 6 (lines 160 - 166), in which we assume that the more significant effect of R2 as compared with R1 is probably due to the fact that R2 substituents are located much closer to the reaction site (methylidene group).

4) There are some format errors in Table 3, in which certain lines are missing and "Me, Me" shouldn't be bold.

Answer

It could be caused by incompatibility of the software. Hopefully, everything is in order now.

Reviewer 4 Report

1) Can the authors explain why they chose carboplatin as the reference compound, given that it has different mechanistic of action with their compound?

2) It would be more convincing and impressive if the authors could test the cytotoxicity of these compounds on some non-cancer to show the selectivity.

3) The mechanistic studies mainly focused on cellular phenotype, how about their molecular mechanism and target? It is better that the authors study their molecular target in this or future work.

Author Response

Answers to Reviewer 4

1) Can the authors explain why they chose carboplatin as the reference compound, given that it has different mechanistic of action with their compound?

Answer

We used carboplatin also in our earlier papers as the reference compound. Actually, carboplatin exerts its action through alkylation of DNA. A well-known γ-lactone parthenolide, has the same mechanism of action (Michael type addition to cellular -SH groups) as our compounds. Since we have tested parthenolide cytotoxicity before, these data have been added to Table 3 for comparison.

2) It would be more convincing and impressive if the authors could test the cytotoxicity of these compounds on some non-cancer to show the selectivity.

Answer

Indeed, testing the best analog 11c on a healthy cell line is planned.

3) The mechanistic studies mainly focused on cellular phenotype, how about their molecular mechanism and target? It is better that the authors study their molecular target in this or future work.

Answer

We agree with the Reviewer on this matter. This is mostly a synthetic study, with only some preliminary biological tests. Further reserch is being planned.